# Specificity of *Loxosceles* α clade phospholipase D enzymes for choline-containing lipids: Role of a conserved aromatic cage

**Emmanuel E. Moutoussamy**[1,2], **Qaiser Waheed**[1,2¤a], **Greta J. Binford**[3], **Hanif M. Khan**[1,2¤b], **Shane M. Moran**[4], **Anna R. Eitel**[4], **Matthew H. J. Cordes**[4], **Nathalie Reuter**[1,5]*

**1** Computational Biology Unit, Department of Informatics, University of Bergen, Bergen, Norway, **2** Department of Biological Sciences, University of Bergen, Bergen, Norway, **3** Department of Biology, Lewis and Clark College, Portland, Oregon, United States, **4** Department of Chemistry and Biochemistry, University of Arizona, Arizona, United States, **5** Department of Chemistry, University of Bergen, Bergen, Norway

¤a Current address: Department of Chemistry, Bioscience and Environmental Engineering, University of Stavanger, Stavanger, Norway
¤b Current address: Centre for Molecular Simulation, Department of Biological Sciences, University of Calgary, Calgary, Alberta, Canada
* nathalie.reuter@uib.no

**Data Availability Statement:** All the MD trajectories are uploaded to the Norwegian national infrastructure for research data (NIRD), have been issued a DOI (10.11582/2021.00099) and can be

## Abstract

Spider venom GDPD-like phospholipases D (*SicTox*) have been identified to be one of the major toxins in recluse spider venom. They are divided into two major clades: the α clade and the β clade. Most α clade toxins present high activity against lipids with choline head groups such as sphingomyelin, while activities in β clade toxins vary and include preference for substrates containing ethanolamine headgroups (*Sicarius terrosus*, St_βIB1). A structural comparison of available structures of phospholipases D (PLDs) reveals a conserved aromatic cage in the α clade. To test the potential influence of the aromatic cage on membrane-lipid specificity we performed molecular dynamics (MD) simulations of the binding of several PLDs onto lipid bilayers containing choline headgroups; two *SicTox* from the α clade, *Loxosceles intermedia* αIA1 (Li_αIA) and *Loxosceles laeta* αIII1 (Ll_αIII1), and one from the β clade, St_βIB1. The simulation results reveal that the aromatic cage captures a choline-headgroup and suggest that the cage plays a major role in lipid specificity. We also simulated an engineered St_βIB1, where we introduced the aromatic cage, and this led to binding with choline-containing lipids. Moreover, a multiple sequence alignment revealed the conservation of the aromatic cage among the α clade PLDs. Here, we confirmed that the i-face of α and β clade PLDs is involved in their binding to choline and ethanolamine-containing bilayers, respectively. Furthermore, our results suggest a major role in choline lipid recognition of the aromatic cage of the α clade PLDs. The MD simulation results are supported by *in vitro* liposome binding assay experiments.

accessed using the following URL: https://archive.sigma2.no/pages/public/datasetDetail.jsf?id=10.11582/2021.00099.

**Funding:** NR and EM acknowledge funding from the Research Council of Norway (Norges Forskningsråd, grants #251247 and #288008). MC acknowledges funding from the National Science Foundation (grant #1808716). The funders had no role in study design, data collection and analysis, decision to publish, or preparation of the manuscript. The Research Council of Norway: https://www.forskningsradet.no/ National Science Foundation, https://www.nsf.gov/.

**Competing interests:** The authors have declared that no competing interests exist.

## Author summary

Envenomation following bites from recluse spiders (*Loxosceles*) causes loxoscelism, a necrotic tissue breakdown in mammals, and leads to skin degeneration and systemic reactions in the worst case. Recluse spiders belong to the Sicariidae family which also includes six-eyed sand spiders in the genera *Sicarius* and *Hexopthalma*. While sicariid spiders are found natively on all continents except Australia, treatments of loxoscelism are typically antibody based and available in some regions of the Americas. Sphingomyelinase D/phospholipase D enzymes are one of the major toxins in venom of sicariid spiders, and have been divided in two clades called α and β. The activity of α and β clades toxins differs; most α clade toxins present high activity against lipids with choline headgroups (-N$(CH_3)_3{}^+$) such as sphingomyelin, while activities in β clade toxins vary and include preference for substrates containing ethanolamine headgroups (-$NH_3{}^+$). When comparing the structures of two α clade toxins and one β clade toxin, we noticed the presence in the α clade toxins only of a cage consisting of three aromatic amino acids. In this work we used numerical molecular simulations to probe the role of this cage in the preference of α clade toxins for choline head groups over ethanolamine head groups.

## Introduction

Sphingomyelinase D/phospholipase D (SMaseD/PLD) enzymes are one of the major toxins in venom of spiders in the family Sicariidae [1–3] including *Loxosceles*, the recluse spiders. Sicariid spiders also include six-eyed sand spiders in the genera *Sicarius* and *Hexopthalma* [4]. The gene family comprising the sicariid SMaseD/PLD toxins is called *SicTox* reflecting evidence that this venom toxin is a synapomorphy for the family Sicariidae and is not a homolog of other PLDs [5]. Envenomation by recluse spiders causes necrotic tissue breakdown in mammals, a condition known as loxoscelism, and leads to skin degeneration and systemic reactions in the worst cases [2]. *SicTox* enzymes are sufficient causative agents for this syndrome in mammals [6] and thus are pharmaceutical targets [7,8]. While sicariid spiders are found natively on all continents except Australia, treatments of loxoscelism are typically antibody based and available in some regions of the Americas [9].

*SicTox* enzymes catalyse the cleavage of lipid headgroups forming an alcohol and a cyclic phospholipid [10]. The structures of two *Loxosceles* PLD have been resolved by X-ray (Fig 1): *Loxosceles laeta* Ll_αIII1 (PDB ids: 1XX1 [11], 2F9R [11]) and *Loxosceles intermedia* Li_αIA1 (PDB id: 3RLH [12]). Their fold consists of a distorted TIM-barrel, a domain also found in many other phospholipases. Their membrane-binding region is called the i-face (for *interfacial* face) and is thought to consist of the loops β2α2 (also called *catalytic* loop), β5α5 and β6α6 (*flexible* loop) [11], all shown on Fig 1.

The *Loxosceles* SMaseD/PLD enzymes are classified in either the α clade or the β clade [5]. Members of the α-clade have high catalytic activity against sphingomyelin (SM), while activity in the β clade is more variable [13–18]. Only β clade enzymes have been found in *Sicarius* and *Hexopthalma* venoms while *Loxosceles* contain both α and β clade *SicTox* toxins paralogs. Using $^{31}$P NMR, Lajoie *et al.* recently showed that a β clade enzyme from *Sicarius terrosus* (St_βIB1) had a strong preference for substrates with ethanolamine headgroups vs. choline-containing lipids [13] in detergent micelles containing the lipid substrates. Lajoie et *al.* also measured the activity of an α and a β clade enzyme from *Loxosceles arizonica* La_αIB2bi and La_β1D1. La_αIB2bi presents a high activity against choline-containing lipids and interestingly La_βID1 from the β clade shows little discrimination between ethanolamine and

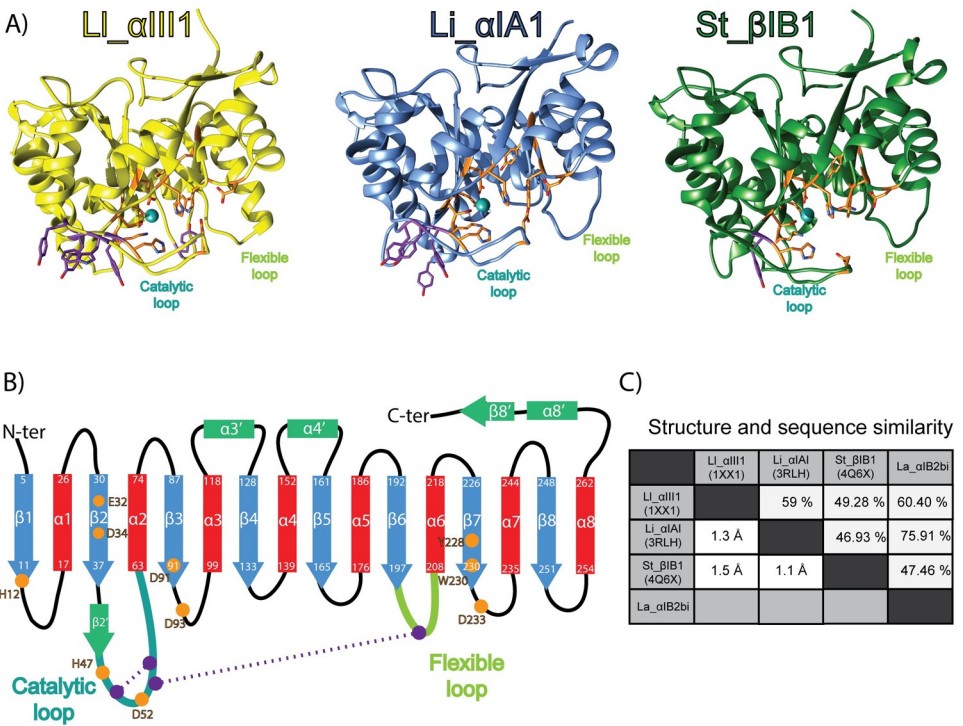

**Fig 1. Overview of the SicTox PLD structures.** A) Structures of phospholipases D used in this work. Li_αIII1, Li_αIA1 and St_βIB1 are represented in yellow, blue and green, respectively. The side chains of amino acids critical for substrate binding and catalysis are shown in orange. The aromatic residues located at the putative i-face are indicated in purple. B) PLD topology: strands (blue) and helices (red) of canonical $(\alpha/\beta)_8$ barrel, other secondary structure elements (green), catalytic loop (cyan), flexible loop (green), disulphide bridges (purple). Residue numbering is based on the Li_αIA1 structure (PDB id: 3RLH). C) Structure and sequence similarity: pairwise RMSD (lower left triangle) and sequence similarity (upper right part).

choline-containing substrates. The substrate specificity of the SMases is thought to be related to the relative abundance of ethanolamine-containing lipids in the cell membranes of arthropods (19), which are preferred prey of sicariid spiders [19]. Lajoie *et al.* also resolved the X-ray structure of St_βIB1 (PDB ID 4Q6X, Fig 1); it resembles the structures of the α clade *SicTox* enzymes from *Loxosceles* and there are only subtle differences in the active or ligand binding sites. This led the authors to suggest that the i-face of the *SicTox* SMases/PLDs is likely to play a role in specific lipid recognition.

While the three previously mentioned loops are thought to constitute the i-face, there has not been any experimental investigation to identify which particular amino acids are involved in membrane binding. As demonstrated by many examples, molecular simulations have proven to be a useful tool to predict interfacial binding sites (IBS) of peripheral membrane proteins and map protein-lipid interactions [20–24]. Of particular relevance are our efforts at characterizing the membrane binding site of a bacterial phosphatidylinositol-specific phospholipase C from *Bacillus thurigensis* (*Bt*PI-PLC) [20,24,25]. *Bt*PI-PLC binds preferentially to PC-rich lipid bilayers, is folded as a distorted αβ TIM barrel, and just as for the *SicTox* SMases/PLDs the *Bt*PI-PLC IBS consists of a few loops located on the same side of the αβ-barrel as the active site. Using a combination of MD simulations and *in vitro* $K_d$ determination of the wild type enzyme and mutants, we could show that the preference of *Bt*PI-PLC for PC lipids was the result of cation-π interactions between surface-exposed tyrosine residues and the choline

groups of PC lipids [20,24–26] and that those cation-π interactions between choline and interfacial tyrosine could contribute up to 2 to 3 kcal/mol to the protein-membrane affinity [27].

Closely observing the amino acid sequences and available X-ray structures of the *Sicarius* St_βIB1 and the two α-clade *Loxosceles* enzymes Ll_αIII1 and Li_αIA1, we noticed that the i-face of the α-clade enzymes is enriched in tyrosine and tryptophan residues. Li_αIA1 counts three tyrosines on the catalytic loop (Y44, Y46, Y60), Ll_αIII1 has three tyrosines (Y44, Y46, Y62) and a tryptophan (W60), while St_βIB1 has only one tyrosine on this loop (Y44). In addition, Li_αIA1 has a tyrosine (Y98) on the β3α3 loop and Ll_αIII1 has a tyrosine (Y169) on β5α5. Interestingly, the catalytic loop of La_αIB2bi also has three tyrosines and a tryptophan at positions equivalent to Y44, Y46, Y62 and W60 in Ll_αIII1, according to a sequence alignment [13]. The position and orientation of these conserved aromatic amino acids in the X-ray structures of Ll_αIII1 and Li_αIA1 (Fig 1) is compatible with the formation of an aromatic cage as observed in proteins binding choline-containing ligands [28][26,29][30]. We pose the hypothesis that the tyrosine and tryptophan residues on the i-face of La_αIB2bi and other α-clade enzymes provide a mechanism to selectively recognize choline-containing lipids as ligands.

In this work, we evaluate this hypothesis using MD simulations and liposome binding assay experiments. As there is no X-ray structure of La_αIB2bi we chose to simulate Ll_αIII1 and Li_αIA1 instead. We identify the amino acids of the i-face involved in membrane binding by characterizing their interactions with lipid bilayers containing either only palmitoyl-oleyl-phosphatidyl choline lipids (POPC) or a mixture of POPC, sphingomyelin (SM) and cholesterol (CHOL). We also seek to understand how St_βIB1 interacts with phosphatidylethanolamine (PE) lipids and performed simulations of the β clade enzyme with bilayers consisting of both PC and PE lipids. Finally, we introduced aromatic amino acids in selected positions of the β clade enzyme to evaluate if it then acquires affinity towards choline-containing bilayers. We also performed *in vitro* liposome binding assay to evaluate the capacity of La_αIB2bi, wild type (WT) St_β1B1 and R44Y/S60Y St_β1B1 to bind to SM:CHOL (1,1) liposomes.

## Methods

### Dataset for structural and multiple sequence alignment

We retrieved X-ray structures from the Protein Data Bank (PDB) [31] for Li_αIA1 (PDB id: 3RLH [12]), Ll_αIII1 (PDB id: 1XX1 ([11], chain A) and St_βIB1 (PDB id: 4Q6X [13]). The three structures were aligned with MUSTANG (v 3.2.3) [32]. Clustal Omega [33] was used to align multiple sequences. Default parameters were used. We retrieved the sequences of 25 PLDs representatives of subgroups in the two clades (12 α- and 6 β-clade PLDs). Their UNI-PROT identifiers are: P0CE83 (Li_αIA2ai), P0CE80 (Li_αIA1a), P0CE81 (Li_αIA1bi), Q1W695 (Li_αII1), P0DM60 (Li_αIVA1), C0JB04 (Lru_αIC2), A4USB4 (Li_αV1), C0JAX7 (Ld_αIB3ai), P0CE79 (Lr_αIA1ii), Q5YD75 (Lr_αIB1), Q5YD77 (Lb_αIB1a), Q4ZFU2 (La_αIB2bi), Q8I914 (Ll_αIII1), Q1KY79 (Ll_αIII2), K9USW8 (Lg_αIC1), A0A0D4WV12 (StβIB1), Q8I912 (Ll_βIA1), Q2XQ09 (Li_βIA1i), Q1W694 (Li_βID1), C0JB41 (Lsp_βIE2i), C0JB55 (Sd_βIF1), C0JB68 (Spa_βIIA1), C0JB92 (Lsp_βIII1), Q5YD76 (Lb_βIA1a) and A0A0D4WTV1 (La_βID1).

### System setup

Lipid bilayers. Bilayers consisting of 256 lipid molecules (128 lipids in each leaflet) were built with the CHARMM-GUI [34] using the Membrane Builder module [35–37]. POPC and POPC:POPE (50:50) bilayers were simulated for 200 ns (NPT) while the POPC:PSM:CHOL (70:20:10) bilayer was simulated for 500 ns (NPT) using the CHARMM36 force field [38–40] The calculated area per lipid for POPC is 65.0 Å$^2$ in agreement with earlier simulations using

**Table 1. Composition and size of the simulated systems.** Each was simulated twice for 300 ns (production run), and one replica of the simulation of St_βIB1 on the POPC bilayer was extended to one μs. The raw trajectories are available at the NIRD Research Data archive (DOI: 10.11582/2021.00099).

| System | Protein | Lipid bilayer | Total number of atoms | Number of water molecules | Ions | Box dimensions x/y/z (Å) |
|---|---|---|---|---|---|---|
| 1 | Li_αIA1 | 256 POPC | 106131 | 22514 | 1 Cl⁻ 1 Mg²⁺ | 89/89/130 |
| 2 | Ll_αIII1 | 256 POPC | 106757 | 22663 | 4 Na⁺ 1 Mg²⁺ | 89/89/130 |
| 3 | St_βIB1 | 256 POPC | 114556 | 25277 | 3 Na⁺ 1 Mg²⁺ | 89/89/130 |
| 4 | St_βIB1 R44Y/S60Y | 256 POPC | 106572 | 22613 | 4 Na⁺ 1 Mg²⁺ | 89/89/130 |
| 5 | Li_αIA1 | 50 PSM 180 POPC 26 CHOL | 105028 | 22783 | 1 Cl⁻ 1 Mg²⁺ | 85/85/135 |
| 6 | Ll_αIII1 | 50 PSM 180 POPC 26 CHOL | 105081 | 22741 | 4 Na⁺ 1 Mg²⁺ | 85/85/135 |
| 7 | St_βIB1 | 50 PSM 180 POPC 26 CHOL | 112307 | 25164 | 3 Na⁺ 1 Mg²⁺ | 85/85/135 |
| 8 | St_βIB1 | 128 POPC 128 POPE | 104812 | 22413 | 3 Na⁺ 1 Mg²⁺ | 88/88/132 |
| 9 | Li_αIA1 | 128 POPC 128 POPE | 112718 | 25094 | 1 Cl⁻ 1 Mg²⁺ | 88/88/142 |

the same force field (65.5 Å$^2$) [21,38]. The experimental value reported by Kučerka is 68.5 Å$^2$ [41]. The addition of POPE to construct a POPC:POPE (50:50) bilayer decreases the calculated area per lipid to 61.8 Å$^2$, in good agreement with the 59.4 Å$^2$ from recent simulation data [42]. As expected [43], the addition of PSM and CHOL to POPC also reduces the area per lipid, measured to be 52.5 Å$^2$ for the POPC:PSM:CHOL (70:20:10) mixture.

Protein-bilayer systems. We set up nine protein-bilayer systems: Li_αIA1 (PDB id: 3RLH [12]) and St_βIB1 (PDB ID 4Q6X [13]) each with a POPC, a POPC:PSM:CHOL (70:20:10), and a POPC:POPE (50:50) bilayer, and Ll_αIII1 (PDB ids: 1XX1 [11]) with a POPC and a POPC:PSM:CHOL (70:20:10) bilayer. Note that the X-ray structures include a magnesium ion in the active site. In addition a mutant of St_βIB1 was prepared with a POPC bilayer; two amino acids (R44 and S60) were replaced by tyrosines using UCSF Chimera [44]. Protein structures were placed slightly above the surface of the relevant bilayers at a distance of 48 Å between the center of mass of bilayer and the center of mass of the protein (corresponding to a minimum protein-bilayer distance between 4 and 6 Å depending on the protein). The proteins were positioned in the binding orientation predicted by OPM [45] for Li_αIA1 and Ll_αIII1. The initial configuration of St_βIB1 WT and mutants are the binding orientation predicted by the PPM server [45]. The systems were solvated and neutralized using VMD (v 1.9.1) [46]. The pKas of ionizable side chains were predicted using PROPKA3 [47,48] and none indicated a deviation from the standard protonation states of individual amino acids at pH 7. The composition of the nine simulated systems, included number of lipids, water molecules and ions, is reported in Table 1.

## Molecular dynamics simulations

All simulations were performed using NAMD (v 2.13) [49] with the CHARMM36 force field [38–40] and its CHARMM-WYF extension for the treatment of aromatics-choline interactions [50,51]. While the ability of additive molecular mechanics force fields to faithfully model

cation−π interactions is not obvious given the nature of those interactions [52], the CHARMM-WYF extension has been thoroughly validated for phosphatidylcholine-aromatics interactions [50,51]. The systems were first subjected to energy minimization with conjugate gradients (10000 steps) and an equilibration of 2 ns in the NPT ensemble. Next the production runs were performed in the same ensemble for 300 ns using the coordinates and velocities of the last step of the equilibration run. All the steps were done with an integration step of 2 fs. We ran two replicates for each of the systems listed in Table 1. Replicas started from the same minimized structure but independent equilibration steps were performed for each replica using different velocity distributions, the velocity distributions were generated using different seeds for the random number generator. The equilibration run of each replica was followed by a production run (see above for details on production runs). One replica of the simulation of St_βIB1 on the POPC bilayer was extended to 1 μs. Before equilibration and production runs, the mutant was subjected to extra equilibration in water for 1 ns. The temperature was set to 310 K and controlled using Langevin dynamics (temperature damping coefficient: 1.0). The pressure was set to 1 atm using the Langevin piston method with an oscillation period of 200 fs and a damping timescale of 50 fs. During each simulation the ratio of the unit cell in the x-y plane was maintained constant (use Constant Ratio keyword in NAMD). The SHAKE algorithm was applied to constrain all bonds between hydrogen and heavy atoms, including those in water molecules to keep water molecules rigid [53]. Particle mesh Ewald (PME) was used for the treatment of long range electrostatic interactions [54]. A Lennard-Jones switching function (10–12 Å) was used for van der Waals interactions. Simulation frames were saved at a frequency of 10ps. The NB-fix corrections from CHARMM36 were used for the ions [55]. All simulations were uploaded to the Norwegian national infrastructure for research data (NIRD, https://archive.norstore.no/) and can be accessed using the following DOI: 10.11582/2021. 00099.

## Trajectory analysis

The analysis of the trajectories was done on the window 150–300 ns, since all proteins were bound to the bilayers and the proteins RMSD were stable during that part of the production runs (Figs A-I in S5 Text). Trajectory analysis was done using CHARMM (v33b1) [56] for the inventory of lipid-protein interactions, VMD (v 1.9.1) [57] for the electron density plots and MD Analysis [58,59] for the calculation of the depth of insertion. We analysed three types of interactions: hydrophobic contacts, hydrogen bonds and cation-π interactions which we identified using the same definition as in Grauffel *et al*. [24]. Briefly hydrophobic contacts are considered to exist if two non-bonded candidate atoms are within a distance of 3 Å. Candidate atoms are atoms in aliphatic groups of amino acids side chains (list provided in supporting information, Table A in S1 Text). For hydrogen bonds the distance between the hydrogen and the hydrogen bond acceptor should be below or equal to 2.4 Å, and the angle between the hydrogen bond donor, the hydrogen and the hydrogen bond acceptor should be higher than or equal to 130˚. Cation−π interactions between the aromatic rings of tyrosines and tryptophans were considered to exist when all distances between the aromatic carbon and the choline nitrogen were below 7 Å. In addition, these distances should not differ by more than 1.5 Å. Interactions were considered only if the criteria were met for at least 10 ps and if the interaction was observed in both replicates. The depth of insertion was calculated for the last frame of each simulation trajectory; we define the depth of insertion for a given amino acid in a given simulation frame as the distance between its Cα atom and the average upper phosphate plane. The distance between the aromatic cage (or a single aromatic amino acid) and the plane of the choline nitrogen atoms was also calculated when relevant. It corresponds to the distance

between the center of mass of the aromatic rings involved in the aromatic cage (Y44, Y46, Y60) (or the COM of the aromatic amino acid) and the average plane defined by the position of the choline nitrogen atoms. The plane and aromatics COM were defined at every frame. We also calculated the angle between the P-N vector and the normal to the membrane for POPC lipids bound to the proteins, and for POPC not bound to the proteins. This was done for systems 1, 2, 5, 6 and 9, to analyse the headgroup orientation.

## Liposome binding assays

T7 expression vectors (pHIS8) containing genes encoding La_αIB2bi and St_β1B1 were constructed previously [19,13]. The R44Y/S60Y variant of St_β1B1 was constructed using Quik Change mutagenesis (Agilent, Santa Clara, California, USA) of the wild-type construct. La_αIB2bi, St_β1B1 and R44Y/S60Y St_β1B1 proteins were then expressed and purified by nickel affinity chromatography as described previously [10], followed by extensive dialysis into insect physiological saline (IPS; 5 mM potassium phosphate [pH 6.5], 100 mM potassium chloride, 4 mM sodium chloride, 15 mM magnesium chloride, 2 mM calcium chloride). Thin lipid films were prepared by dissolving equimolar amounts of egg sphingomyelin and cholesterol to a total lipid concentration of 9 mg/mL in 2:1 (v/v) chloroform:methanol, evaporating the solvent in a glass test tube with a weak stream of argon while rotating the tube, and drying the film under vacuum at 45˚C for at least 2 h. Films were converted to liposomes (large multilamellar vesicles) by suspending them at 36 mg/mL lipid concentration in 10 mM HEPES (pH 7.5), 150 mM sodium chloride, followed by heating at 65˚C for 1 h with occasional vortexing. For the liposome binding assay, 1.5–2 µg protein in 10 µL IPS was mixed with 10 µL of an 18 mg/mL liposome solution and incubated at 25˚C for 30 min, followed by centrifugation for 10 min at ambient temperature at 16000 *g*. Supernatant and pellet were separated, converted to gel samples of equal volume, loaded onto Tris-tricine SDS-PAGE gels, and electrophoresed for at least 45 min at 150 V. Gels were stained with Coomassie Brilliant Blue R-250, destained, and photographed. Bands were quantified using ImageJ software [61]. Values reported in the text are the average of three experiments; reported errors represent the standard error of the mean.

## Results

### Choline-aromatics cation-π interactions between the aromatic cage in α clade enzymes and POPC lipids

We performed molecular dynamics simulations of each of the three PLDs (Li_αIA1, Ll_αIII1 and St_βIB1) in the presence of two different bilayers: POPC and POPC:PSM:CHOL (70:20:10). Each simulation was initiated with the protein positioned slightly above the bilayer. The RMSD of the protein backbone with respect to structures after equilibration remains low during the production runs (Cf Figs A and B in S5 Text) and show deviations of at most 1.2 Å.

The β clade *Sicarius* enzyme St_βIB1 does not bind to the PC-containing bilayers within the same timeframe as the α clade enzymes (300 ns) and diffuses away rapidly (Figs 2 and 3A), in agreement with its low affinity for POPC lipids. We verified that on an extended 1µs-long simulation of St_βIB1.The enzyme approached the bilayer at several occasions (Fig D in S6 Text) but did not reach a stable orientation, as illustrated by the large range of variations of the distance between Y46 of the catalytic loop and the bilayer (Fig E in S6 Text).

The α clade *Loxosceles* enzymes, on the other hand, Li_αIA1 and Ll_αIII1, bind to bilayers composed of POPC (Figs 2 and 3A and 3B) and POPC:SM:CHOL (Figs 3A and 3C and S1) within 100 ns. Fig 3A shows the electron density profiles (EDP) and Fig 3D the depth of insertion below the phosphate plane calculated for each amino acid and the POPC bilayer. The two

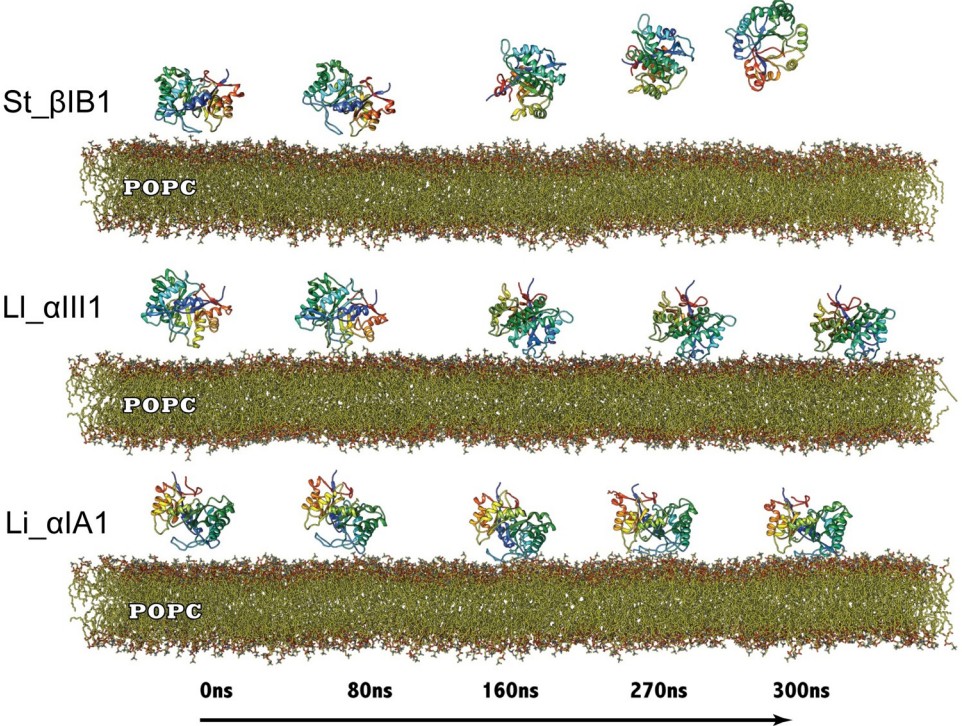

**Fig 2. Simulation trajectory snapshots of Li_αIA1, Ll_αIII1 and St_βID1 on a POPC bilayer.** Snapshots were taken at time t = 0, 80, 160, 270 and 300 ns of the production run.

α clade enzymes are anchored rather superficially in the bilayer, irrespective of the type of bilayer, and only a few residues are inserted under the phosphate plane (Fig 3D). Both Li_αIA1 and Ll_αIII1 interact with the lipids through their β2α2 catalytic loop while the other loops, including β6α6 remain higher above the interface (Fig 3B–3D). We observed hydrophobic contacts with the lipid chains for only a few residues; I49 and P50 from Li_αIA1 and I58 from Ll_αIII1 (Table 2).

Amino acids involved in hydrogen bonds, hydrophobic contacts and cation-π interactions with the lipids are listed in Table 2. Numerous residues located in the catalytic loop establish a hydrogen bond network with the bilayer. Most occupancies are low (<50%) and most hydrogen bonds are achieved by the side chains except for I49, C51, G54 in Li_αIA1. We observe more hydrogen bonds in Li_αIA1 than Ll_αIII1 but even in Li_αIA1 the occupancies of hydrogen bonds are not high except for K58 (above 70%). The side chain of R59 in Ll_αIII1 also engages in long-lasting hydrogen bonds with the lipids (> 90% occupancy). Noteworthy are the hydrogen bonds achieved by Y44, Y46, Y60 and Y98 in Li_αIA1 and Y62 from Ll_αIII1.

These tyrosines also establish cation-π interactions with the choline groups of the lipid heads (Table 2 and Fig 4). In Li_αIA1 Y44, Y46 and Y60 form an aromatic cage occupied by a choline group. The occupancies of the cation-π interaction between those tyrosines and the bilayer are between 29 and 46%. In Ll_αIII1, Y44, Y46, W60 form an equivalent cage and the occupancies of the cation-π interactions are between 28 and 89%. In the course of the 300 ns of simulation, the cage can be occupied by either the same lipid or by up to 3 subsequent lipids during the simulation (Figs F-H in S6 Text). Moreover, the two α clade PLDs contain an additional tyrosine establishing cation-π interactions with the bilayer: Y98 from the β3α3 loop in

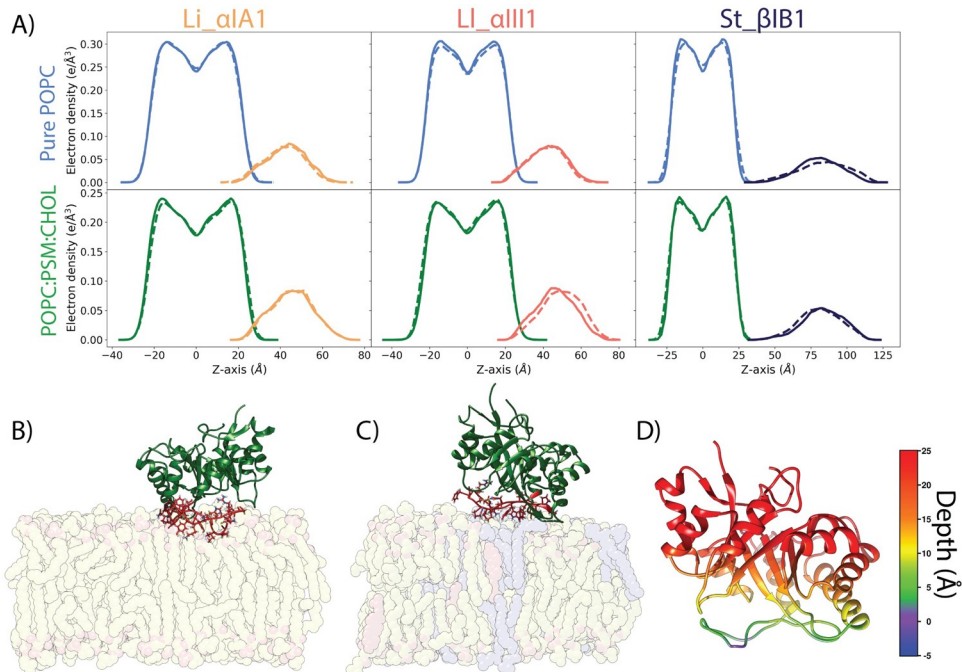

**Fig 3. Li_αIA1 and Ll_αIII1 bound to POPC and POPC:PSM:CHOL (70:20:10).** A) Electron density plots for Li_αIA1, Ll_αIII1 and St_βIB1 on the POPC bilayer (top row) or on the POPC:PSM:CHOL bilayer (bottom row), replica 1 (solid lines) and replica 2 (dashed lines). The densities for the POPC and POPC:PSM:CHOL bilayers are represented in light blue and green, respectively. The density distributions for the proteins are represented in orange (Li_αIA1), pink (Ll_αIII1) and dark blue (St_βIB1). B) Bound orientation of Li_αIA1 on a pure POPC bilayer. C) Binding orientation of Ll_αIII1 on a POPC:PSM:CHOL bilayer. D) Depth of anchorage of Li_αIA1 into the POPC bilayer. The depth is calculated for each amino acid as the distance between its C-alpha and the phosphate plane in the last frame of simulation (relevant values are given in Tables A and B in S2 Text).

Li_αIA1 (46–60% depending on the bilayer) and Y62 from the catalytic loop in Ll_αIII1 (39–46%).

## Dense hydrogen bond network between aspartates in the β clade enzyme and POPE

Lajoie *et al.* showed that St_βIB1 turns over ethanolamine-containing ceramides and lysophospholipids 2 to 3 orders of magnitude faster than corresponding choline-containing lipids. The binding pocket for the lipid headgroup in PLDs has been identified using molecular docking [13] and does not appear to account for the difference of lipid specificity. We ran simulations of St_βIB1 in the presence of a POPC:POPE (50:50) bilayer to predict the binding site of the *Sicarius* enzyme and analyse the protein-lipid interactions.

Unlike the simulations of St_βIB1 on the POPC and POPC:SM:CHOL bilayers (Fig 3) the enzyme binds rapidly to the POPC:POPE bilayer (Fig 5A). The density plots (Fig 5B) show an anchoring slightly deeper than for the α-clade enzymes in the PC-containing bilayers (Fig 3A, top panel). This is also visible from the insertion depth at the end of the simulation which is deeper for catalytic loop (β2α2) and the flexible loop (β6α6) of St_βIB1 (Fig 5D and Table C in S2 Text) than for the corresponding loops of the α-clade enzymes (Fig 3D and Table A in S2 Text). The orientation with respect to the bilayer surface is also slightly different with the flexible loop β6α6 being involved in interactions with the lipids (Fig 5C and 5D). This is reflected by the number of hydrophobic contacts and hydrogen bonds (Table 3). As a result of a deeper

**Table 2. Inventory of interactions between Li_αIA1 (3RLH) or Ll_αIII1 (1XX1) and two types of bilayers: POPC and POPC:PSM:CHOL.** Hydrophobic contacts are given as average number of contacts per frame. Hydrogen bonds and cation-π occupancies are shown in percentages. Residues in bold indicate hydrogen bonds involving the amino acid backbone. Asterisk (*) indicates strictly conserved positions between Li_αIA1 and Ll_αIII1.

| SSE | residue | Li_αIA1 | | | | Ll_αIII1 | | | |
|---|---|---|---|---|---|---|---|---|---|
| | | Bilayer A | | Bilayer B | | Bilayer A | | Bilayer B | |
| | | R1 | R2 | R1 | R2 | R1 | R2 | R1 | R2 |
| | | Hydrogen bonds (%) | | | | | | | |
| β2α2 | K38 | - | - | - | - | 33.6 | 36.5 | 40.0 | 36.7 |
| | Y44* | 15.5 | 20.8 | 24.9 | 22.0 | - | 15.0 | 24.3 | 16.9 |
| | Y46* | 68.0 | 63.4 | 50.3 | 50.3 | - | - | - | - |
| | **I49** | 23.2 | 21.9 | 20.1 | 21.7 | - | - | - | - |
| | **C51*** | 45.9 | 40.8 | 38.6 | 35.6 | - | - | - | - |
| | **G54** | 20.0 | 26.4 | 35.9 | 35.8 | - | - | - | - |
| | K58 | 72.5 | 79.9 | 83.2 | 77.4 | - | - | - | - |
| | R59 | - | - | - | - | 91.4 | 89.4 | 100.0 | 92.4 |
| | W60 | - | - | - | - | 15.4 | 11.0 | 15.0 | 20.6 |
| | Y62 | - | - | - | - | 50.3 | 34.8 | 45.0 | 46.6 |
| β3α3 | Y98 | 37.0 | 46.9 | 35.9 | 32.9 | - | - | - | - |
| | | Hydrophobic contacts (avg. contact per frame) | | | | | | | |
| β2α2 | I49 | 4.0 | 4.2 | 5.0 | 4.0 | - | - | - | - |
| | P50* | 3.0 | 3.4 | 4.3 | 3.2 | - | - | - | - |
| | I58 | - | - | - | - | 1.0 | 0.6 | 0.6 | 0.7 |
| | | cation-π interactions (%) | | | | | | | |
| β2α2 | Y44* | 35.6 | 28.9 | 30.5 | 34.8 | 28.8 | 27.5 | 34.9 | 29.6 |
| | Y46* | 40.4 | 46.0 | 43.8 | 44.3 | 73.9 | 67.9 | 84.2 | 86.3 |
| | Y60/W60 | 44.7 | 44.8 | 40.6 | 33.2 | 84.4 | 80.3 | 83.1 | 89.0 |
| | Y62 | - | - | - | - | 41.2 | 45.9 | 39.4 | 41.7 |
| β3α3 | Y98 | 50.5 | 60.6 | 46.6 | 47.9 | - | - | - | - |

anchoring at the interface we observe more hydrophobic contacts with the lipid tails than between the α clade enzymes and PC bilayers. Amino acids of the highly conserved region of the flexible loop (I194, T195, L198, P199) all have on average more than 1.0 hydrophobic contacts per frame with the lipid tails in both replicas. The phenylalanine at position 54, which corresponds to a glycine in the two α clade enzymes, has a high count of hydrophobic contacts too.

We observe a dense hydrogen bond network between the lipids and several loops of St_βIB1 (Table 3). In addition to the catalytic (β2α2) and flexible (β6α6) loops, the β7α7 loop and the amino acids of the α8 helix engage in long-lasting hydrogen bonds with lipid headgroups (occupancies between 50% and 100%). Loop β5α5 and helix α7 engage only in weak hydrogen bonds. Most of the hydrogen bonds involve the lipid phosphate group. However, six residues establish hydrogen bonds specifically with ammonium groups of the ethanolamine headgroups: N169, D201, D202, D229, E235 and E259 (Table 3). Among those D201 and D202 establish long-lasting hydrogen bonds with POPE ammonium groups (Fig 6). The hydrogen bonds between other amino acids–N169, D229, E235, E259 –and POPE headgroups have a relatively lower occupancy (< 40% of the simulation time). Interestingly in the two α clade enzymes, amino acids at position 202 are not acids but instead there is a leucine (L202) in Li_αIA1 and a methionine (M202) in Ll_αIII1. Position 201 is an aspartic acid (D201) in Ll_αIII1 but a glycine (G201) in Li_αIA1.

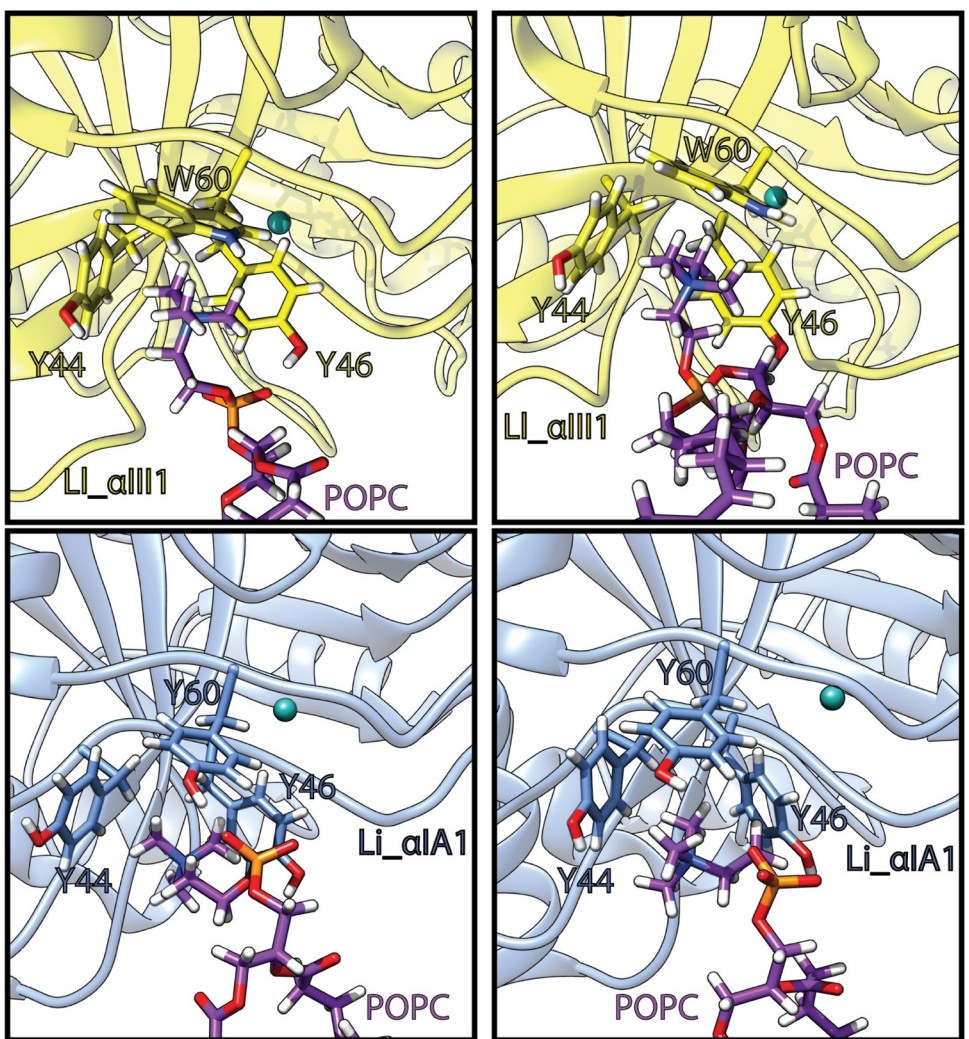

**Fig 4. Phosphatidylcholine headgroups trapped in the aromatic cage of Li_αIA1 (blue) and Ll_αIII1 (yellow).**
Lipids carbon atoms are in purple, the magnesium ion is represented as a cyan ball. The snapshots are extracted from
the last 150ns of the simulation trajectories.

## Specific recognition of PC headgroups by the α clade enzyme Li_αIA1

In order to evaluate the selectivity of the aromatic cage in α-clade PLD for PC headgroups, we
performed simulations of Li_αIA1 on the POPE:POPC (50:50) bilayer. In the first replica, the
protein remained relatively close to the bilayer (Fig C in S6 Text) but did not achieve stable
binding (Fig F in S6 Text). In the second replica, Li_αIA1 did bind to the bilayer after ca. 150
ns (Fig 7A) with the same orientation as on a pure POPC bilayer (Fig 7C) but not as deep (Fig
7B and 7C) as it did in POPC (Fig 3A and 3D). As a consequence we did not observe hydro-
phobic interactions between Li_αIA1 and the POPC:POPE bilayer. The hydrogen bond net-
work between Li_αIA1 and the bilayer is restricted to only six residues and the highest
occupancy is only 48.8% (N56 and POPE lipids). The three tyrosines Y44, Y46 and Y60 are
also involved in cation-π interactions with POPC choline groups but never with POPE lipids
(Fig 7D). The occupancies are reported in Table 4 (between 41 and 92%, Table 4). Y98 also
interacts with POPC via cation-π interactions but has a low occupancy (20%).

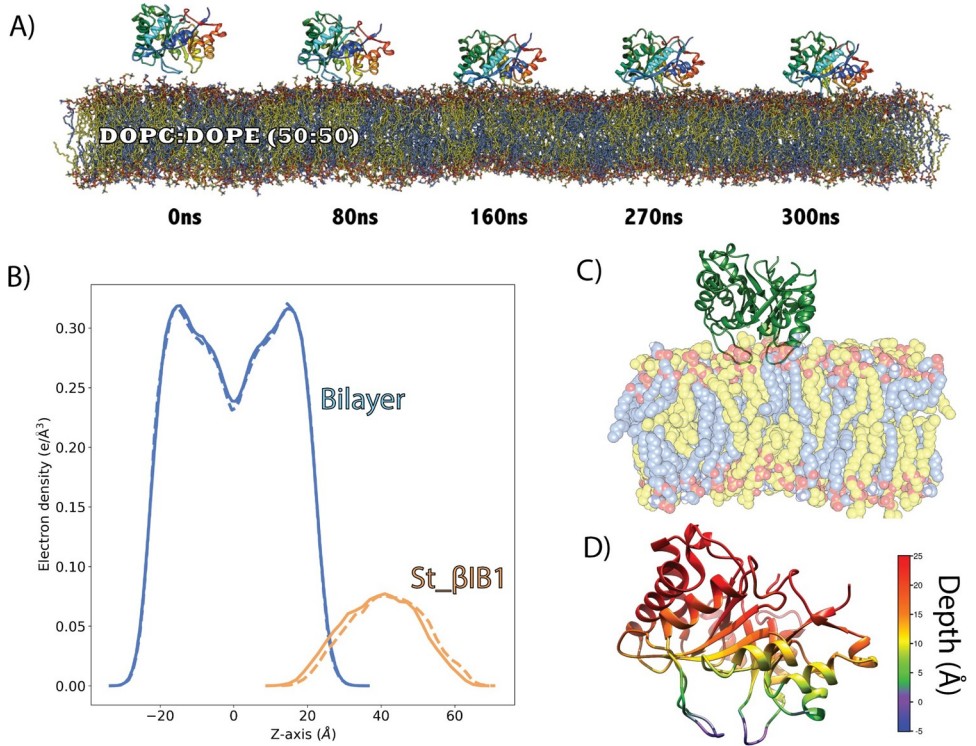

**Fig 5. Binding of St_βIB1 to a POPC:POPE (50:50) bilayer.** A) Simulation snapshots. B) Electron density plot for two replica (solid lines and dashed lines). C) Bound form. D) Depth of anchorage. The depth is calculated for each amino acid as the distance between its C-alpha and the phosphate plane in the last frame of simulation (relevant values are given in Table C in S2 Text).

## Engineering the aromatic cage in St_βIB1 recovers binding to pure POPC bilayer

The aromatic cage present in the α clade Li_αIA1 and Ll_αIII1 but not in St_βIB1, a β clade enzyme, is likely to be important for the recognition of choline groups of PC and SM lipids. We engineered such an aromatic box on St_βIB1 by substituting R44 and S60 by tyrosines and simulated the resulting structure on a POPC bilayer. We checked that the mutations did not alter the protein structure. During the simulations we observed no major structural differences compared to the X-ray structure of the wild-type (S2 Fig). The backbone RMSD between the starting structure and the R44Y/S60Y double mutant oscillates between 1.0 Å and 1.2 Å during the production run, and the RMSD between WT St_βIB1 X-ray structure and R44Y/S60Y is equal to 1 Å at the end of the simulation. Unlike what we observed with the WT enzyme, and similar to Li_αIA1 and Ll_αIII1, the double mutant binds quickly to the POPC bilayer (Fig 8). The interaction with lipids happens mostly at the catalytic loop (Fig 8B and 8C and Table 5). The aromatic cage interacts with choline head groups and unlike for the wild type enzyme, few hydrogen bonds and hydrophobic contacts are established with the interfacial lipid groups (Table 5). This is in line with the shallow insertion of the enzyme. Overall, the interactions follow the same pattern as that obtained for Li_αIA1 with the POPC bilayer.

## Liposome binding assay

Binding assays with mixed sphingomyelin:cholesterol (SM:CHOL) liposomes (Fig 9) provide some experimental support for the aromatic cage effect found in the simulations. La_αIB2bi,

**Table 3. Inventory of interactions between WT St_βIB1 and a POPC:POPE bilayer.** Hydrophobic contacts are given as average number of contacts per frame. Hydrogen bonds and cation-π occupancies are reported as percentages. Residues in bold indicate hydrogen bonds involving the amino acid backbone. Underlined residue names indicate hydrogen bonds involving the ethanolamine headgroup.

| SSE | Residue | R1 | R2 |
|---|---|---|---|
| | **Hydrogen bonds (%)** | | |
| β2α2 | **C51** | 99.8 | 57.9 |
| | **C53** | 17.2 | 19.7 |
| | S56 | 99.7 | 99.8 |
| | T58 | 40.6 | 41.8 |
| | R59 | 32.2 | 22.1 |
| β5α5 | N169 | 29.8 | 12.9 |
| β6α6 | R200 | 12.4 | 49.3 |
| | D201 | 80.5 | 85.4 |
| | D202 | 58.9 | 65.7 |
| β7α7 | D229 | 29.0 | 13.1 |
| | K230 | 68.2 | 60.6 |
| α7 | **E231** | 33.5 | 36.3 |
| | S232 | 56.4 | 37.5 |
| | S233 | 40.5 | 40.0 |
| | E235 | 40.8 | 34.4 |
| α8 | Y249 | 76.1 | 84.8 |
| | R252 | 85.4 | 81.5 |
| | E259 | 27.4 | 32.5 |
| | R260 | 30.5 | 36.2 |
| | **Hydrophobic contacts (avg. contact per frame)** | | |
| β2α2 | P50 | 0.1 | 0.1 |
| | C51 | 1.2 | 0.8 |
| | C53 | 0.7 | 0.4 |
| | F54 | 1.4 | 1.3 |
| | R55 | 1.0 | 0.8 |
| β6α6 | I194 | 1.0 | 1.1 |
| | T195 | 1.1 | 1.2 |
| | C197 | 0.7 | 0.6 |
| | L198 | 1.5 | 1.7 |
| | P199 | 1.6 | 1.5 |
| α7 | K230 | 1.3 | 1.2 |

an α clade protein possessing an aromatic cage and known to prefer substrates with choline head groups [13], is bound completely by 1:1 SM:CHOL liposomes. The β clade enzyme St_β1B1, which lacks an aromatic cage, is bound poorly, with an average of only 15 ± 4% of the protein sample pulled down by the liposomes relative to a negative control. The aromatic cage variant R44Y/S60Y St_β1B1 showed higher but still incomplete binding, with 28 ± 4% pulled down.

## The aromatic cage is mostly conserved in α clade *SicTox* PLDs

A multiple sequence alignment (MSA) of 25 PLDs from Lajoie *et al.* 2015 [13] and 7 additional samples that represent known diversity (list provided in methods section) shows that the sequences are fairly similar with the lowest sequence identity being around 37% (Li_αIA2ai vs

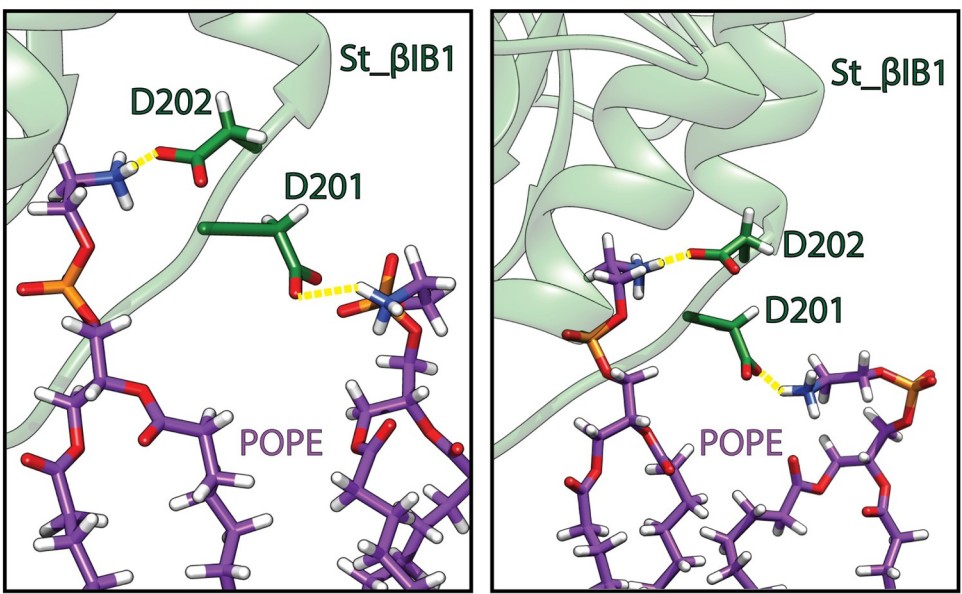

**Fig 6. Hydrogen bonds (yellow) between residues D201 and D202 of St_βID1 and phosphatidylethanolamine headgroups: POPE lipid (carbon atoms in purple), protein (green).** The two snapshots are taken during the last 50 ns of simulation.

Lsp_βIE2i) (Fig 10B). Fig 10A shows the MSA in the region of the catalytic loop (from position 31 to 62, S3 Fig) and of the flexible loop (from position 192 to 204, S3 Fig). The full MSA is given in supporting information (S3 Fig).

The aromatic cage is mostly conserved in the α clade but the complete cage is mostly absent from the β clade enzymes. Y44 and Y46 are strictly conserved in the α clade; position 60 can be either a tyrosine or a tryptophan except for Lg_αIC1 where it is a serine. The cage is not present in Lru_αIC2. In the β clade, only Y46 is mostly conserved (except in Li_βID1 and La_βID1) but position 44 is a charged amino acid (E, R or K) and position 60 is a polar amino acid (S, T or Q). Position 62 is an aromatic (Y) in about half of the α clade PLDs of our dataset, forming an aromatic cage of four residues. In the β clade sequences, this position is not occupied by aromatic residues. Only two representants of the β clade exhibit the aromatic cage (Lsp_βIII1 and Sd_βIF).

In the flexible loop, only two β clade enzymes present an aspartate at position 201 (G, I or V for the other β clade PLDs and G or D for the α clade PLDs). D202 is conserved among the β clade PLDs in our dataset (only two β clade PLDs present a methionine at this position). Interestingly, this position is occupied by a hydrophobic residue in the α clade PLDs (M or L). D201 and D202 are located in the β6α6 loop (flexible loop). The C-terminus side of this loop is richer in aspartate in β clade PLDs than in the α clade PLDs towards the end of this loop (Fig 10).

## Discussion and conclusions

The structures of Li_αIA1 and Ll_αIII1 reveal an aromatic cage formed by amino acids at positions 44, 46 and 60, but this cage is not present in the *Sicarius* PLD (St_βIB1) structure, leading to the hypothesis that this cage is part of the mechanism by which the α clade enzymes achieve high affinity for choline-containing lipid headgroups. We discuss below how our results support this hypothesis.

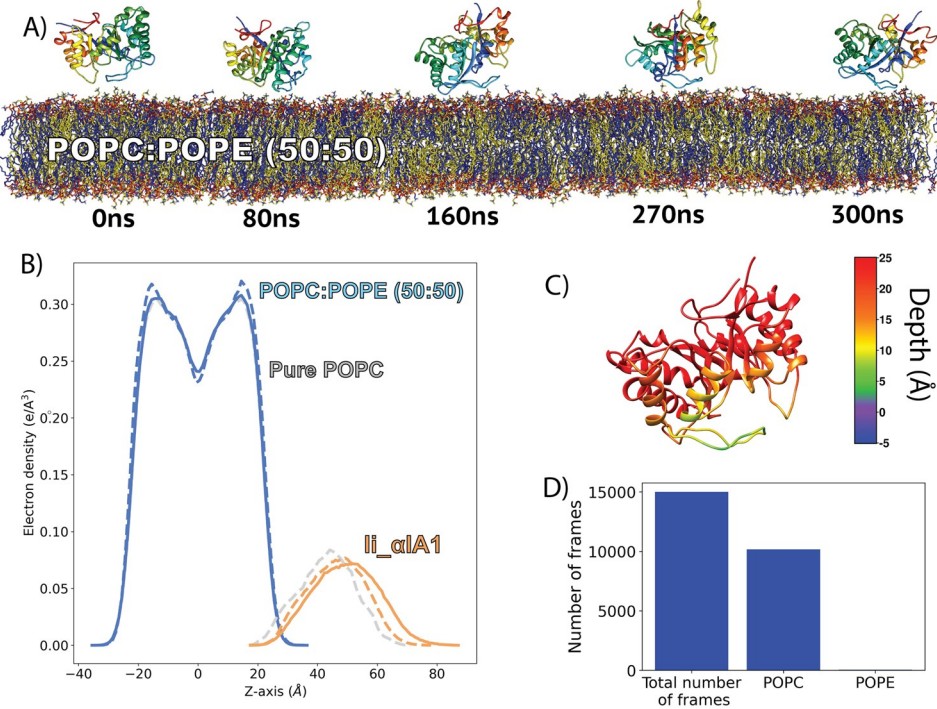

**Fig 7. Binding of Li_αIA1 to a POPC:POPE bilayer.** A) Simulation snapshots for replica. 2. B) Electron density plot for Li_αIA1 and the POPC:POPE (50:50) bilayer for two replicas (solid lines for R1 and dashed lines for R2, blue for the bilayer and orange for the protein). For the sake of comparison, the electron density plots for Li_αIA1 and the pure POPC bilayer are represented in grey. C) Depth of anchorage. D) The number of frames where a POPC or POPE lipid is in the aromatic cage, compared to the total number of frames. A lipid is considered in the cage when the distance between the center of mass of the aromatic cage and the nitrogen atom of the lipid (POPC or POPC) is lower than or equal to 6 Å.

MD simulations showed that Li_αIA1 and Ll_αIII1 bind to choline-containing bilayers with the same binding orientation whereas St_βIB1 remained in solvent. The i-face of the two α clade enzymes on choline-containing bilayers involves the catalytic loop and cation-π

**Table 4. Inventory of interactions between Li_αIA1 (3RLH) and a POPC:POPE bilayer.** Hydrogen bonds and cation-π occupancies are reported as percentages. Residues in bold indicate hydrogen bonds involving the amino acid backbone. All numbers are for interactions with POPC lipids, except those marked with an asterisk (*) which indicates interactions involving the headgroup of POPE lipids. K58 and K59 interact with both POPC and POPE headgroups.

| SSE | Residues | R2 |
|---|---|---|
| **Hydrogen bonds (%)** | | |
| β2α2 | Y46 | 31.4 |
| | **G54** | 36.0 |
| | N56 | 48.8* |
| | K58 | 43.0/25.3* |
| | K59 | 39.0/41.0* |
| | Y60 | 20.0 |
| **cation-π interactions (%)** | | |
| β2α2 | Y44 | 41.3 |
| | Y46 | 47.0 |
| | Y60 | 92.0 |
| β3α3 | Y98 | 20.2 |

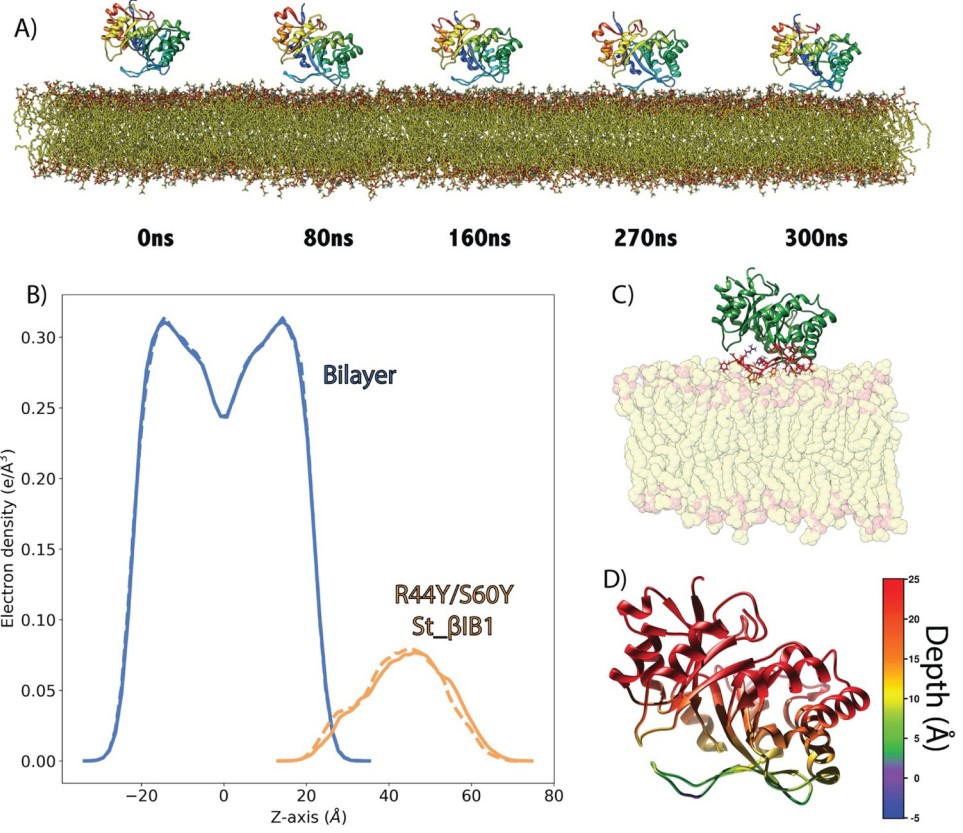

**Fig 8. Binding of R44Y/S60Y St_βIB1 to a POPC Bilayer.** A) Simulation snapshots. B) Electron density plot for R44Y/S60Y St_βIB1 and the POPC bilayer for two replica (solid lines and dashed lines). C) Bound protein orientation. D) Depth of anchorage. The depth is calculated for each amino acid as the distance between its C-alpha and the phosphate plane in the last frame of simulation (relevant values are given in Table A in S2 Text).

**Table 5. Inventory of interactions between R44Y/S60Y St_βIB1 and a POPC bilayer.** Hydrophobic contacts are given as average number of contacts per frame. Hydrogen bonds and cation-π occupancies are shown in percentages. Residues in bold indicate hydrogen bonds involving the amino acid backbone.

| SSE | Residues | R1 | R2 |
|---|---|---|---|
| **Hydrogen bonds (%)** | | | |
| β2α2 | K38 | 39.1 | 68.5 |
| | Y44 | 10.5 | 20.8 |
| | Y46 | 60.0 | 54.3 |
| | **C51** | 53.3 | 67.3 |
| | S56 | 46.7 | 58.3 |
| | **C57** | 20.6 | 10.3 |
| | Y60 | 20.0 | 35.8 |
| **Hydrophobic contacts (avg. contact per frame)** | | | |
| β2α2 | V49 | 3.0 | 4.3 |
| | P50 | 2.5 | 1.9 |
| **cation-π interactions (%)** | | | |
| β2α2 | Y44 | 43.6 | 39.9 |
| | Y46 | 35.6 | 40.2 |
| | Y60 | 44.8 | 20.3 |

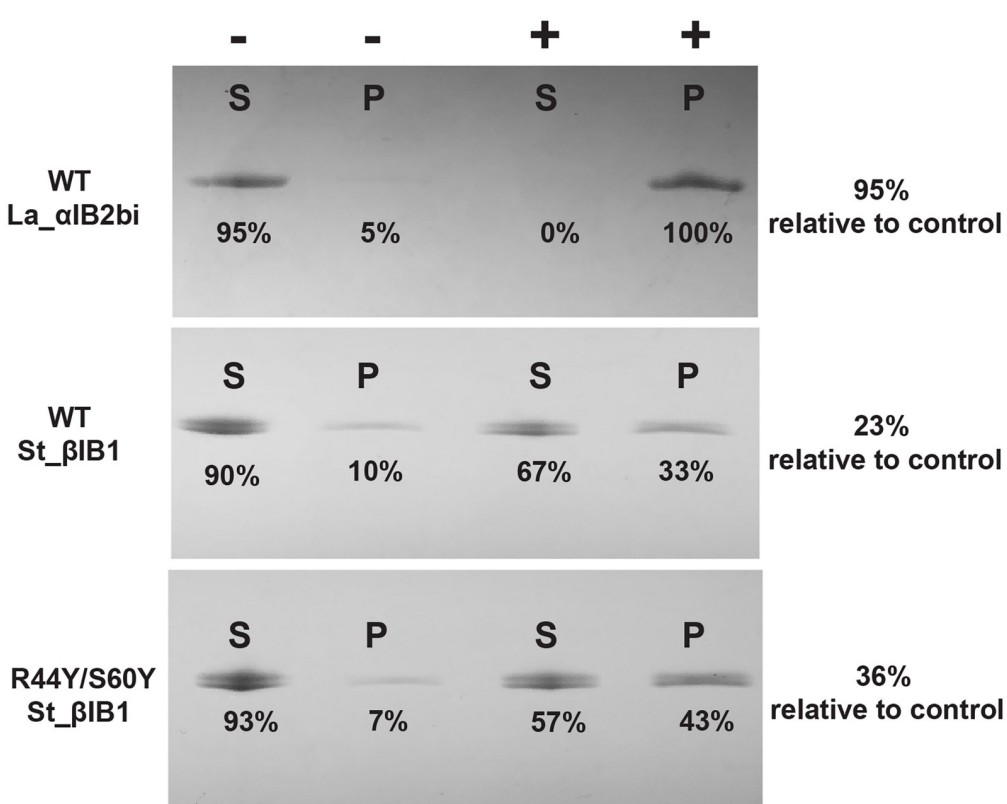

**Fig 9. Liposome binding assay for WT La_αIB2bi, WT St_βIB1 and R44Y/S60Y St_βIB1 on SM:CHOL (1:1) liposomes.**
P and S refer to pellet and supernatant, respectively. The "-" symbol indicates the control lanes (without liposomes) and the
"+" symbol the lanes with the protein and the liposomes. The St-βIB1 and mutant data shown here represent the single run
closest to the mean increase. Data for the three individual runs are found in Table A in S3 Text. Across three experiments the
mutant R44Y/S60Y St_βIB1 showed 10–17% higher levels of pulldown than the WT, giving a mean increase of 13 ± 2%.

interactions between the aromatic residues forming the cage and choline headgroup. The aromatic residues located at the i-face are engaged in strong cation-π interactions with choline headgroups. The liposome binding assays show that La_αIB2bi, an α clade enzyme with the aromatic cage, shows strong binding to SM:CHOL (1:1) liposomes whereas St_ βIB1 binds poorly to those.

The structure of the aromatic cage observed during simulations resembles aromatic cages found in other proteins around choline groups [26,29,62–64]. Such cages consist of 2 to 4 aromatic side chains, either only tyrosines as in Li_αIA1, or a combination of tyrosines and tryptophans as we observe Ll_αIII1. Roughly half the sequences in our datasets have a cage consisting of three aromatics and half have a cage of four. The aromatic amino acids arrange as a box forming a hydrophobic binding site for the methylated ammonium whose positive charge is also stabilized by the π-system of the aromatic groups. The shortest distances between carbons of the aromatic ring and methyl groups are within 4 Å. Examples of cages around choline-containing lipids include the structure of the human phosphatidylcholine transfer protein (PDB IDs 1LN1 and 1LN3) complexed with 1,2-dilinoleoyl-*sn*-glycerol-3-phosphorylcholine (DLPC,18:2(9,12)-18:2(9,12)) or 1-palmitoyl,2-linoleoyl-*sn*-glycerol-3-phosphorylcholine (PLPC, 16:0–18:2(9,12)) bound to a cage consisting of three tyrosines and a tryptophan. In the engineered N254Y/H258Y *Sa*PI-PLC two tyrosines and one tryptophan bind a choline ion

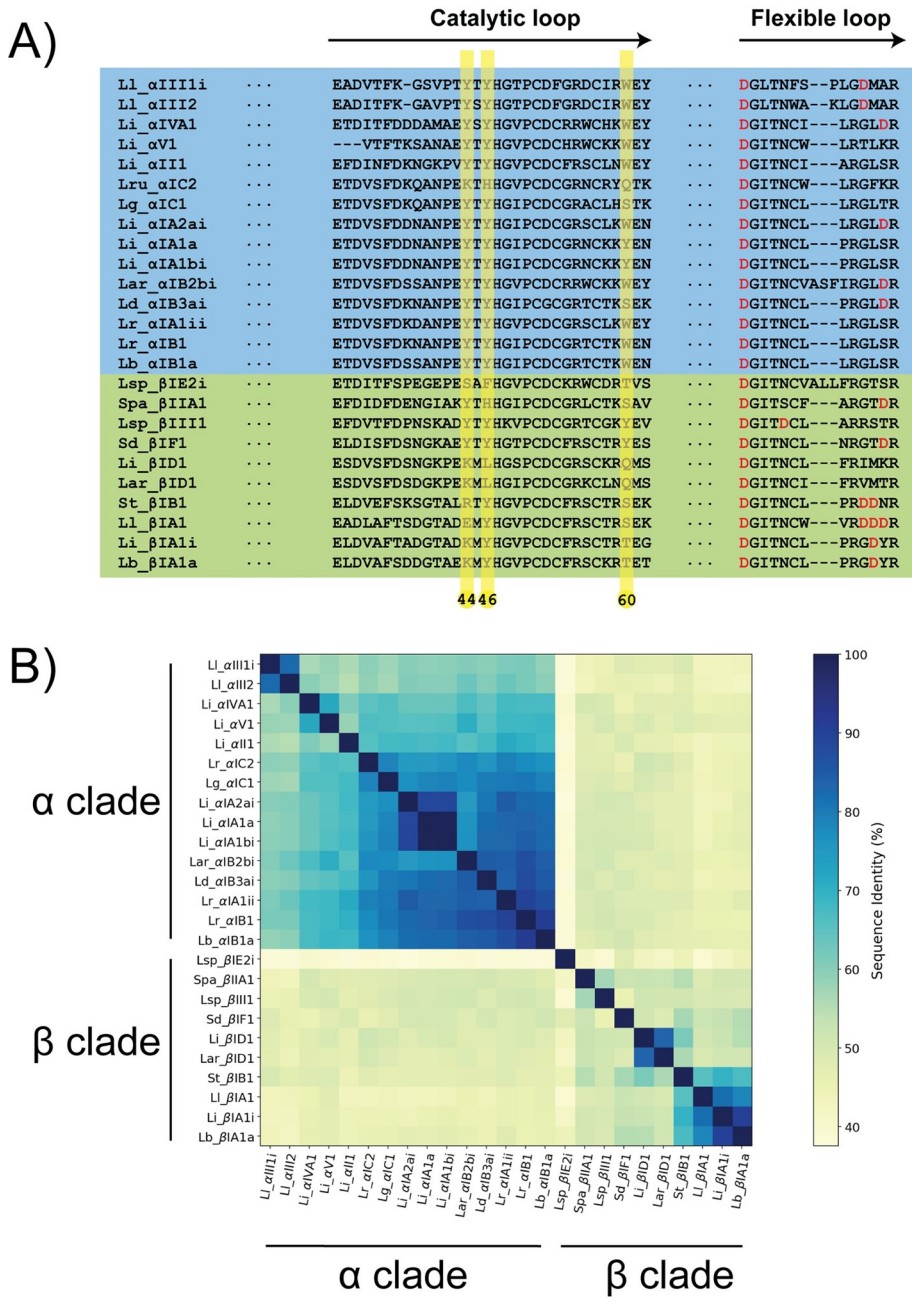

**Fig 10. Amino acid sequences similarities in the catalytic and flexible loops.** A) Multiple sequence alignment of the catalytic and the flexible loop. The full sequence alignment is given as supporting information (Fig H in S5 Text). B) Sequence similarity matrix.

(PDB ID 4I90) or the choline group of diC4PC (PDB ID 4I9J) [26]. Interestingly, in our simulations, the orientation of the bound PC lipids headgroups do not significantly differ from that of PC groups in the bilayer (Table A in S4 Text) but without a thorough evaluation of the CHARMM-WYF force field for caged POPC lipids, we cannot rule out the inverse conformational selection model proposed by *Bacle et al.* [65].

Further, using simulations of Li_αIA1 on a POPC:POPE (50:50) bilayer we could show that the aromatic cage preferably binds choline headgroup over ethanolamine headgroups. This

result is consistent with the observation made by Cheng et al. [26]. Indeed, when the engineered *Sa*PI-PLC was incubated with PC containing vesicles (PG/PC (0.2 mM/0.8 mM)), they observed that 68% of the proteins bound to the vesicles. By contrast, only 36% of the proteins bound to the bilayer when the vesicles contained PE lipid (PG/PE (0.2 mM/0.8 mM)). The aromatic cage is a lipid recognition motif able to discriminate between choline-containing lipids and ethanolamine-containing lipids.

Inspired by the work of Cheng et al., we engineered an aromatic cage in the β clade enzyme *in silico* [26]. Starting from *Staphylococcus aureus* PI-PLC (*Sa*PI-PLC) which does not bind to PC-rich vesicles, Cheng et al. increased its affinity for PC-rich liposomes by engineering an aromatic cage inspired by the sequence and structure of the PC-specific homologue *Bt*PI-PLC. X-ray structures of the modified *Sa*PI-PLC enzyme soaked with phosphocholine or DiC$_7$PC (1,2-Diheptanoyl-sn-Glycero-3-Phosphocholine) revealed that the choline groups were positioned in the aromatic cage. Our simulations of the engineered St_βIB1 (R44Y/S60Y St_βIB1) lead to a stable interaction with a POPC bilayer, unlike the results we obtained with the wild-type. Just as we observed for the α clade enzymes, a choline headgroup was observed residing in the aromatic cage during most of the simulation time. The liposome binding assay of R44Y/S60Y St_βIB1on SM:CHOL (1:1) liposomes shows significantly increased binding compared to WT St_βIB1. This suggests that the aromatic cage is one important factor for the binding of membranes rich in choline head groups; on the other hand, the lack of complete pulldown suggests that other factors must also contribute to the weak binding of St_βIB1 to choline-rich membranes, relative to α clade enzymes such as La_αIB2bi. We do not see obvious indications from the simulations of α-clade members binding to choline-rich bilayers of what other key interactions might be missing in St_βIB1.

An alignment of the sequences of 25 representative PLDs from different subgroups of the two clades shows that the aromatic cage is highly conserved in the PLDs from the α clade and not present in the well characterized PLDs from the β clade. Overall, the conservation of a cage in most α-clade enzymes, and its positive role in choline recognition based on simulations and *in silico* and *in vitro* mutagenesis data, support a significant role for the cage in targeting α-clade toxins to choline-rich membranes. Other things being equal, this targeting should increase access of the α-clade enzymes to the choline-containing substrates SM and lysophosphatidylcholine (LPC), toward which they have generally high activity. This likely contributes to the efficacy of α-clade toxins in triggering loxoscelism in humans, but presumably also plays an adaptive role in prey capture, the principal biological function of *Loxosceles* venom [19]. Meanwhile, St_βIB1, a β clade paralog that strongly prefers substrates with ethanolamine head groups, lacks the cage, as do most other β clade enzymes. In both simulation and experiment, its binding to choline-rich membranes is increased by introduction of the cage. One caveat is that La_βID1, a second β clade enzyme that also lacks the cage, has fairly high activity against choline-containing substrates, though low preference relative to ethanolamine-containing substrates [13] suggesting that other determinants of head group preference must also contribute.

To better understand the specificity of St_βIB1 for PE-rich membrane, we also ran simulations of WT St_βIB1 in the presence of a POPC:POPE (50:50) bilayer. As opposed to the simulation on bilayers without POPE lipids, WT St_βIB1 bound to the bilayer with an i-face composed of the catalytic (β2α2) and flexible (β6α6) loops, as well as loop β7α7 and helix α8. WT St_βIB1 is anchored slightly deeper in the bilayer than the α-clade enzymes and in lieu of cation−π interactions, it engages in a dense hydrogen bond network with the lipid phosphates and headgroups. Interestingly two aspartates located on the flexible loop establish hydrogen bonds with the ethanolamine headgroup. Early assessments of sequence diversity in the MSA indicate that the C-terminus region of the flexible loop is richer in aspartate in the β clade than in the α clade.

Beyond suggesting the molecular basis for the specificity of α-clade PLDs for choline-rich membranes, this work is relevant for the design of drugs targeting PLDs from sicariid spiders. We suggest that the aromatic cage might be worth targeting to weaken protein-membrane interactions of the α clade enzymes. Last but not least the role of the evolutionary conserved aromatic cage in *SicTox* PLD enzymes, together with already existing evidence of interfacial choline-aromatics cation-π interactions in other protein families [23,24,63,66], further confirms that such interactions could be a general mechanism for specific recognition of the surface of eukaryotic cells or other choline-rich biological membranes.

## Supporting information

**S1 Fig. Snapshots from simulations of Li_αIA1, Ll_αIII1 and St_βIB1 with a POPC:PSM: CHOL (70:20:10) bilayer.** Snapshots at 0, 80, 160, 270 and 300 ns were extracted to follow the evolution of the systems during the simulations.
(TIF)

**S2 Fig. Structural alignment between WT St_βIB1 and R44Y/S60Y St_βIB1.** The structures used for the alignment are final frames of each production run of WT St_βIB1 and R44Y/S60Y St_βIB1 each on a pure POPC bilayer. The calculated RMSD of the protein backbone is 1 Å in both cases.
(TIF)

**S3 Fig. Full multiple sequences alignment of α and β clade SicTox PLDs.** The Uniprot identifiers of the aligned sequences are provided in the Methods section of the article. The green background highlights the catalytic and the flexible loops. The proteins simulated in this study are indicated with a yellow star. The red numbering corresponds to the numbering of the St_βIB1 structure (PDBid: 4Q6X). The black numbering indicates the position in the multiple sequence alignment.
(TIF)

**S1 Text. Inventory of the candidate atoms for the analysis of hydrophobic interactions.**
(PDF)

**S2 Text. Depth of Anchoring.**
(PDF)

**S3 Text. Liposome binding assay data.**
(PDF)

**S4 Text. Conformations of POPC headgroups.**
(PDF)

**S5 Text. Stability of the molecular dynamics simulations.**
(PDF)

**S6 Text. Protein-bilayer distances during the simulations.**
(PDF)

## Author Contributions

**Conceptualization:** Emmanuel E. Moutoussamy, Greta J. Binford, Hanif M. Khan, Matthew H. J. Cordes, Nathalie Reuter.

**Formal analysis:** Emmanuel E. Moutoussamy.

**Funding acquisition:** Matthew H. J. Cordes, Nathalie Reuter.

**Investigation:** Qaiser Waheed, Shane M. Moran, Anna R. Eitel.

**Methodology:** Emmanuel E. Moutoussamy, Qaiser Waheed, Greta J. Binford, Matthew H. J. Cordes, Nathalie Reuter.

**Project administration:** Matthew H. J. Cordes, Nathalie Reuter.

**Resources:** Matthew H. J. Cordes, Nathalie Reuter.

**Software:** Emmanuel E. Moutoussamy, Qaiser Waheed.

**Supervision:** Matthew H. J. Cordes, Nathalie Reuter.

**Validation:** Matthew H. J. Cordes, Nathalie Reuter.

**Writing – original draft:** Emmanuel E. Moutoussamy.

**Writing – review & editing:** Emmanuel E. Moutoussamy, Greta J. Binford, Hanif M. Khan, Matthew H. J. Cordes, Nathalie Reuter.

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
