## [Decision Letter · Decision Letter 0]

21 Sep 2021

Dear Prof. Reuter,

Thank you very much for submitting your manuscript "Specificity of Loxosceles α clade phospholipase D enzymes for choline-containing lipids: role of a conserved aromatic cage" for consideration at PLOS Computational Biology.

As with all papers reviewed by the journal, your manuscript was reviewed by members of the editorial board and by several independent reviewers. In light of the reviews (below this email), we would like to invite the resubmission of a significantly-revised version that takes into account the reviewers' comments.

We cannot make any decision about publication until we have seen the revised manuscript and your response to the reviewers' comments. Your revised manuscript is also likely to be sent to reviewers for further evaluation.

Sincerely,

Guanghong Wei

Associate Editor

PLOS Computational Biology

Nir Ben-Tal

Deputy Editor

PLOS Computational Biology

Reviewer's Responses to Questions

**Comments to the Authors:**

Reviewer #1: This manuscript presents mainly a set of molecular dynamics (MD) simulations of spider venom PLD structures focusing on the aromatic cage and interactions with the PC headgroup. Comparisons are made with a protein in the b clade (St_bIB1) and how it interacts with the membrane when lacking this aromatic cage. Overall, this work is interesting but there lacks proof of equilibration of membrane-protein interaction and statistics associated with the experimental work. Details on this and other aspects are provided below.

General Issues:

1. Membrane-protein equilibration: The authors focus on a simple metric of RMSD of protein structure but do not focus on presenting a key metric associated with the proteins binding to the membrane. There needs to be some quantitative proof that the proposed bound states are near equilibrium. Key plots of distance of the protein relative to a metric in the membrane is a start. Moreover, plots of block averages of protein density profiles in comparison with the membrane is important. The concern is that the binding of peripheral membrane proteins can be more on the microsecond timescale and either long simulations or enhanced sampling is typically needed to probe stable binding.

2. St_bIB1 with POPC: The authors base low affinity on how this protein doesn’t bind to the POPC membranes but experimental with SM/Chol membrane suggests some binding. It appears that the timescale might not be enough (300ns) to probe binding for these PC headgroups. Have you tested this binding with longer simulations or with enhanced sampling?

3. Liposome Binding Assay: The binding assay is relatively crude and the figure lacks statistical information and the text on page 8 suggests some variance but unclear how this was obtain or if this is standard error or standard deviation. Quantifying intensity of bands with numerical values is qualitative at best with this method. Can one statistically state that the mutated St_bIB1 shows more binding? The band intensity suggest maybe but is this reproducible and do you have the same amount of mutated protein in comparison with the wildtype?

4. St_bIB1 other factors in binding: The liposomal assay suggests other factors are contributing to weak binding beyond the lack of an aromatic cage. The discussion lacks potential details on what the other factors might be. Can simulations provide some insight?

Specific Comments

Abstract: Please define PLD.

Ionizable residues: Do you have ionizable residue with this protein? Was a pKa estimate made on the protein to verify the proper states of these residues?

Initial protein setup: The authors provide a center of mass distance describing the placement of the protein relative to the center of the bilayer. This is not that informative and should be supplemented with a measure of minimal distance with the protein or some other distance related to key membrane binding domain/s on the protein.

Figure 3A: The density profiles are not described in the figure or figure captions. I can figure this out but the protein vs. membrane density should be clearly stated.

Cutoff LJ: The authors forgot to mention the cutoff method and length used in these simulations.

Reviewer #2: Review uploaded as an attachment

Reviewer #3: Moutoussamy et al. set out to study a hypothesis that a key reason why a certain set of phospholipases (α-clade) has a higher affinity towards phosphatidylcholine (PC) lipid headgroups than a related set (β-clade), is that α-clade typically contains a structural element dubbed 'aromatic cage' (comprising 2 to 4 tryptophans or tyrosines), whereas β-clade typically lacks such a cage.

They find support for their hypothesis primarily by performing and analysing unbiased all-atom molecular dynamics (MD) simulations; a liposome-binding assay provides additional experimental support.

I find the work reasonable, and aptly executed, and thus likely to be suitable to be published in PLOS Computational Biology. I would, however, like to draw the authors' attention to the following points:

(1) In the Introduction (p5) the authors spell out their hypothesis as:

"[T]he tyrosine and tryptophan residues on the i-face of La_αIB2bi and other α-clade enzymes provides a mechanism to selectively recognize choline-containing lipids as ligands."

However, it appears that none of the performed MD simulations with the aromatic-cage-containing enzymes had other than PC headgroups in their membranes. Therefore, it is not possible to say if the aromatic cage (composed of the said tyrosine and tryptophan residues on the i-face) recognizes SELECTIVELY the PC headgroups. To test the selective recognition, simulations of membranes containing (also) some other headgroup (such as PE) should be performed. If the stated hypothesis holds true, the α-clade enzymes (or more specifically the aromatic cage in them) would not recognize these other headgroups—or would at least bind them much less frequently than the PC headgroups.

(2) The abstract states that: "Here, we confirmed the membrane binding site of α and β clade PLDs on choline and ethanolamine-containing bilayers, respectively." The statement is acceptable concerning the α-clade—although of course 'confirmed' is a bit strong word for a predominantly MD simulation study—where the binding site (for PC) appears to be the aromatic cage; however, it is not quite clear to me what the confirmed binding site for the β clade on ethanolamine is. Could the authors please clarify?

(3) What is the conformation of the PC headgroup bound in the aromatic cage? In particular, is the conformation something typically seen in the other (free) PC heads? Or does the conformation adapt to the binding site, as recently claimed by the proponents of the so-called 'Inverse Conformational Selection Model' [Bacle et al, JACS 143 13701 (2021), https://doi.org/10.1021/jacs.1c05549]?

(4) Please make the raw simulation trajectories available on an open data service. A nice option is using the CERN-run Zenodo (zenodo.org), which is free to use, allows trajectories up to 50 gigabytes, and provides DOIs, which one can then cite directly in the manuscript.

(5) When discussing the system preparation, please (i) mention roughly the size (x/y/z) of the simulation box; (ii) provide the Table S1 in the actual paper; and (iii) include in this table the DOIs for raw MD trajectories, see the previous point.

(6) It is stated that the two replicas "differ by the equilibration step". What is meant by this, that is, how do they differ exactly?

(7) Concerning the ions, was the NB-fix correction of CHARMM36 used?

(8) What was the saving frequency in the simulations?

(9) Concerning the pressure control, was it isotropic, that is, all box vectors were scaled with the same factor?

(10) Was SHAKE applied also in waters?

(11) When determining the depth of insertion, the reference location (of the the upper phosphate plane) was "calculated on the last frame of the simulation". Why? To me it seems that this could introduce a systematic error, especially if the membrane shifts vertically during the simulation. To this end, the location of the upper phosphate plane should be determined for each frame separately. In fact, most natural would be to center the trajetory before analysis such that the center of mass of the upper phosphate plane stays at zero, then the z-coordinate can be directly interpreted as the depth of insertion.

(12) In addition to the protein backbone RMSD shown now in the SI, please show as a function of simulation time (i) the simulation box area in the bilayer plane, and (ii) the minimum distance between the protein and its nearest image.

(13) Please show also the (maximum) depth of insertion as a function simulation time for all the simulations.

(14) On p12 it is written that the "density plots (Fig. 5B) show an anchoring slightly deeper than for the α-clade enzymes". This is rather hard for the reader to see, so it might be better to give this information as numbers—say, list the time averages of the maximum insertion depths for each enzyme.

(15) In Fig. 7D, please rotate the color bar showing the insertion depth, such that it intuitively matches the snapshot (negative numbers bottom, positive top).

(16) In Fig 8, the top panel says "Lar_aIIB2bi", but the caption "La_αIB2bi".

(17) In Fig 8, what do the percentages stand for? Is it correct that for each experiment the "S" and "P" percentages (out of which only the "P" is shown), will add to 100? If yes, then I think it would be easier for the reader, if also the "S" percentages were written on the plot.

(18) When discussing the Fig. 8 in the text, two numbers are given: 15+/-4% and 28+/-4%. How were these exactly calculated?

(19) Accuracies in tables. The H-bond and cation-pi occupancies are not significant to tenths of promilles; I would expect to see them maximally in the precision of percentages. Same for hydrophobic contacts; probably a hundreth of a contact (as in Table 2) is not really significant?

(20) What is the cyan ball shown in the snapshots of Fig 4? An ion?

(21) Typos:

- p3, 3rd last line: differences in is the active -> differences in the active

- p4, 7th last line: L1_αIII1 has two tyrosines (Y44, Y46, Y62) -> L1_αIII1 has three tyrosines (Y44, Y46, Y62)

- p7, 5th line: The protein were -> The proteins were

- p8, 6th last line: 10ps -> 10 ps

- p10, 10th last line: diffuses away rapidly (Fig. 2 and 3A) -> diffuses away (Fig. 2 and 3A) [The speed of diffusion can not really be reliably determined from two simulations, I would expect?]

- p10, 8th last line: Figure 4A -> Figure 3A

- p10, 7th last line: (EDP) and the depth of insertion below -> (EDP) and Fig. 3D the depth of insertion of Li_αIA1 below

- p10, 6th last line: The two enzymes -> The two α-clade enzymes

- p10, 2nd last line: with the lipids chains -> with the lipid chains

- p11, 9th last line: 27.54 -> 28

- p12, 3rd line: lipids. - The binding -> lipids. The binding

- p12, 4th line: molecular docking and -> molecular docking [CITATION] and

- p13, last line: Fig. 8 -> Fig. 7A

- p16, 10th last line: ammonium which positive -> ammonium whose positive

- p18, 9th last line: MSA indicate that the-terminus -> MSA indicate that the [N?/C?]-terminus

- caption of Table 1: Asterix -> Asterisk

**Have the authors made all data and (if applicable) computational code underlying the findings in their manuscript fully available?**

Reviewer #1: **No: **No deposited input simulation codes and data on liposomal studies lack any data tables for the estimated variance.

Reviewer #2: **No: **The authors state 'All the MD data are stored locally and are available'.

My understanding was the PLoS Comp Biol required simulation trajectories to be made available in a public repository e.g. zenodo.

Reviewer #3: **No: **Authors have the MD data stored locally, and available upon request. The data should be made directly openly available, e.g., using the Zenodo server (zenodo.org).

PLOS authors have the option to publish the peer review history of their article (what does this mean?). If published, this will include your full peer review and any attached files.

Reviewer #1: No

Reviewer #2: No

Reviewer #3: No
---

## [Decision Letter · Decision Letter 1]

20 Dec 2021

Dear Prof. Reuter,

Thank you very much for submitting your manuscript "Specificity of Loxosceles α clade phospholipase D enzymes for choline-containing lipids: role of a conserved aromatic cage" for consideration at PLOS Computational Biology. As with all papers reviewed by the journal, your manuscript was reviewed by members of the editorial board and by several independent reviewers. The reviewers appreciated the attention to an important topic. Based on the reviews, we are likely to accept this manuscript for publication, providing that you properly modify the manuscript according to the comments/suggestions of Reviewer #3.

Sincerely,

Guanghong Wei

Associate Editor

PLOS Computational Biology

Nir Ben-Tal

Deputy Editor

PLOS Computational Biology

[LINK]

Reviewer's Responses to Questions

**Comments to the Authors:**

Reviewer #1: Updated revision addresses my comments

Reviewer #2: The authors have addressed my comments in the revised version of this ms.

Reviewer #3: See attached pdf

**Have the authors made all data and (if applicable) computational code underlying the findings in their manuscript fully available?**

Reviewer #1: Yes

Reviewer #2: Yes

Reviewer #3: Yes

PLOS authors have the option to publish the peer review history of their article (what does this mean?). If published, this will include your full peer review and any attached files.

Reviewer #1: No

Reviewer #2: No

Reviewer #3: No

Figure Files:

Data Requirements:

Reproducibility:

References:

---

## [Decision Letter · Decision Letter 2]

27 Jan 2022

Dear Prof. Reuter,

We are pleased to inform you that your manuscript 'Specificity of Loxosceles α clade phospholipase D enzymes for choline-containing lipids: role of a conserved aromatic cage' has been provisionally accepted for publication in PLOS Computational Biology.

Best regards,

Guanghong Wei

Associate Editor

PLOS Computational Biology

Nir Ben-Tal

Deputy Editor

PLOS Computational Biology

Reviewer's Responses to Questions

**Comments to the Authors:**

Reviewer #3: The Authors have provided careful responses to all the points raised.

**Have the authors made all data and (if applicable) computational code underlying the findings in their manuscript fully available?**

Reviewer #3: Yes

PLOS authors have the option to publish the peer review history of their article (what does this mean?). If published, this will include your full peer review and any attached files.

Reviewer #3: No

---

## [Editor Report · Acceptance letter]

14 Feb 2022

PCOMPBIOL-D-21-01515R2 

Specificity of </i>Loxosceles</i> α clade phospholipase D enzymes for choline-containing lipids: role of a conserved aromatic cage

Dear Dr Reuter,

I am pleased to inform you that your manuscript has been formally accepted for publication in PLOS Computational Biology. Your manuscript is now with our production department and you will be notified of the publication date in due course.

With kind regards,

Olena Szabo
